# An Ionic-Liquid-Imprinted Nanocomposite Adsorbent: Simulation, Kinetics and Thermodynamic Studies of Triclosan Endocrine Disturbing Water Contaminant Removal

**DOI:** 10.3390/molecules27175358

**Published:** 2022-08-23

**Authors:** Imran Ali, Gunel T. Imanova, Hassan M. Albishri, Wael Hamad Alshitari, Marcello Locatelli, Mohammad Nahid Siddiqui, Ahmed M. Hameed

**Affiliations:** 1Department of Chemistry, Jamia Millia Islamia (Central University), New Delhi 110025, India; 2Department of Chemistry, King Abdulaziz University, Jeddah 21589, Saudi Arabia; 3Department of Physical, Mathematical and Technical Sciences, Institute of Radiation Problems, Azerbaijan National Academy of Sciences, AZ 1143 Baku, Azerbaijan; 4Department of Chemistry, College of Science, University of Jeddah, Jeddah 21589, Saudi Arabia; 5Department of Pharmacy, University “G. d’Annunzio” of Chieti-Pescara, Build B, Level 2, Via dei Vestini, 31, 66100 Chieti, Italy; 6Department of Chemistry and IRC Membranes and Water Security, King Fahd University of Petroleum and Minerals (KFUPM), Dhahran 31261, Saudi Arabia; 7Department of Chemistry, Faculty of Applied Sciences, Umm Al-Qura University, Makkah 21955, Saudi Arabia

**Keywords:** ionic liquid nanocomposite, water treatment, endocrine-disturbing triclosan, simulation, thermodynamics, kinetics

## Abstract

The presence of triclosan in water is toxic to human beings, hazardous to the environment and creates side effects and problems because this is an endocrine-disturbing water pollutant. Therefore, there is a great need for the separation of this notorious water pollutant at an effective, economic and eco-friendly level. The interface sorption was achieved on synthesized ionic liquid-based nanocomposites. An N-methyl butyl imidazolium bromide ionic liquid copper oxide nanocomposite was prepared using green methods and characterized by using proper spectroscopic methods. The nanocomposite was used to remove triclosan in water with the best conditions of time 30 min, concentration 100 µg/L, pH 8.0, dose 1.0 g/L and temperature 25 °C, with 90.2 µg/g removal capacity. The results obeyed Langmuir, Temkin and D-Rs isotherms with a first-order kinetic and liquid-film-diffusion kinetic model. The positive entropy value was 0.47 kJ/mol K, while the negative value of enthalpy was −0.11 kJ/mol. The negative values of free energy were −53.18, −74.17 and −76.14 kJ/mol at 20, 25 and 30 °C. These values confirmed exothermic and spontaneous sorption of triclosan. The combined effects of 3D parameters were also discussed. The supramolecular model was developed by simulation and chemical studies and suggested electrovalent bonding between triclosan and N-methyl butyl imidazolium bromide ionic liquid. Finally, this method is assumed as valuable for the elimination of triclosan in water.

## 1. Introduction

Nowadays, new-generation water pollutants are getting more attention for their removal. These are personal-care products, medicinal and pharmaceutical residues. Among many residues, triclosan has been found as a water pollutant in some places in the world [1]. Further, this compound has been reported in some aquatic animals [2,3]. Basically, triclosan is used as an antimicrobial agent with a wide range of activities. It is used as oral medicine in some countries [4]. It is mixed with many personal products during their preparation. Most commonly, triclosan is mixed with sanitizers, soaps, skin creams, etc. [5]. It is also used in houses, hotels and hospitals to maintain hygiene [6,7,8,9,10,11]. In addition, it is also used as an additive in polymer production, such as polyethylene and polyolefin polymers. Consequently, there are great chances of water contamination by triclosan. It is a pollutant of high concern because of its endocrine-disturbing nature. A good value of log K_ow_ of 4.8 octanol–water partition coefficient made it capable to be bioaccumulative in fatty tissues, which is responsible for toxicity. In addition, the hormonal activity of these pollutants is studied in vitro and in vivo (animals) and showed serious effects [12,13,14,15]. These effects may affect human beings. The most notorious effect is that triclosan may enter the human body through the skin [16,17], which may be a dangerous sign for health. Briefly, triclosan is a serious and toxic pollutant and its detailed health hazards are discussed by Olaniyan et al. [1]. Chemically, triclosan is known as 5-Chloro-2-(2,4-dichlorophenoxy)phenol (Figure 1) with C_12_H_7_Cl_3_O_2_ molecular formula and 289.54 g as molecular mass.

Due to the above discussion, it is very clear that there is an urgent need for fast, economic, reproducible and reliable methods for the removal of triclosan in water. Among many materials, nanomaterials are gaining utility in a wide range of applications, including water treatment [18,19,20,21,22]. Some papers are available on the removal of triclosan. Brose et al. [23] defined triclosan removal in influents, effluents and biosolids. Wu et al. [24] defined the subtraction of triclosan using clay and sandy soils with maximum sorption at a high concentration of 0.05 mg/L. In addition, the maximum sorption was at low pH, which is not the pH of natural water. Moreover, the authors reported maximum sorption in days, which is a high time for fast and economic processes. Tonga et al. [25] described the removal of triclosan using biosolid-derived biochar. The authors reported high sorption at low pH. Khori et al. [26] reported the removal of triclosan by using activated carbon prepared from waste biomass. Although the equilibrium achieved was within 20 min, the adsorption capacity was low, i.e., 80.77%. Fard and Barkdoll [27] reported the removal of triclosan on magnetic nanoparticles. The authors reported maximum removal in 47 min. Yi et al. [28] reported the removal of triclosan using core-shell Fe_3_O_4_@COFs nanoparticles. The removal was good but the adsorbent is costly and it is difficult to use on a large scale. Triwiswaraa et al. [29] reported the removal of triclosan by using char obtained from palm kernel. The authors reported maximum sorption from 6 to 12 h, which is a high time period for a fast and economic process. Jiang et al. [30] reported the removal of triclosan on a silica–zeolite sorbent. The authors reported the removal of triclosan at a high concentration of about 2.0 mico mole. It is clear from the above-cited literature that no one can be used to remove this pollutant on a large scale economically. The reason is that almost all methods took a longer time, which is not feasible in natural water treatment on a large scale. Moreover, mostly, methods are applicable at high concentrations of triclosan while it is present at the microgram level in the water. Taking these facts into consideration, a new N-methyl butyl imidazolium bromide ionic-liquid-based copper oxide nanocomposite was prepared using a facile and green method. The nanocomposite was characterized and used for the removal of triclosan. The sorption mechanism was developed by simulation and chemical methods. The results of the findings are discussed in this article.

## 2. Experimental

Details of the chemicals, reagents, instruments used and experimental protocol are given in the Appendix A. However, some important information is provided herein.

### 2.1. Preparation of N-Methyl Butyl Imidazolium Bromide CuO Nanocomposite

Copper nanoparticles were prepared using green methods by utilizing the appropriate amount of copper acetate and *Acacia arabica* leave extract. Thus, 50 g leaves of *Acacia arabica* was dried in shade, ground in the form of water paste and heated in 1000 mL distilled water at 80 °C for 2 h followed by cooling at room temperature and filtration through Whatman filter paper number 1. Next, 250 mL solution of copper acetate of 0.1 M concentration was prepared and the leaf extract (500 mL) was mixed slowly with constant stirring. The pH of the solution was adjusted to 10 by using dilute sodium hydroxide solution. The solution was stirred for 5 h and kept standing undisturbed at room temperature. The solid part of the solution was centrifuged and washed with water following its activation in a furnace at 500 °C for 5 h. The so-formed nanoparticles of CuO were used to prepare the composite with N-methyl butyl imidazolium bromide. Further, 0.5 g CuO nanoparticles was suspended in 100 mL of water and after that, a solution of N-methyl butyl imidazolium bromide in 50 mL of the concentration of 1.0 g/L was added to a suspension of CuO nanoparticles. The suspension was stirred 2–3 h with constant heating. A blackish product was obtained which was washed with hot water and ethanol. Finally, the product was dried in an oven at 100 °C (yield = 99.5%) and used for further sorption studies.

### 2.2. Characterization of N-Methyl Butyl Imidazolium Bromide Nanocomposite

The obtained N-methyl butyl imidazolium bromide ionic liquid copper oxide nanocomposite was characterized by FT-IR, XRD, SEM and TEM spectroscopic techniques. These methods were utilized as per the typical protocol.

### 2.3. Sorption Study

The uptake capability of N-methyl butyl imidazolium bromide ionic liquid copper oxide nanocomposite was determined by the typical measures of batch mode. Several factors were optimized and these were the amount of triclosan, sorbent dose, medium pH, time of contact and temperature. The variables used were 20–120 µg/L triclosan; 0.1–1.5 g/L for N-methyl butyl imidazolium bromide copper oxide nanocomposite dosage; 3–12 medium pHs; time of 5–40 min with a temperature of 20–30 °C. The lasting triclosan was detected by HPLC as described in the Appendix A. The different models are also given in Supporting Information to analyze the results. The equilibrium capacity of sorption was fixed by the following equation.
Q_e_ (µg/g) = (C_i_ − C_e_) × v/m(1)
where C_i_ and C_e_ are starting and equilibrium amounts of triclosan in µg/L whilst m is the weight of N-methyl butyl imidazolium bromide copper oxide nanocomposite in g/L. The solution volume (v) was in milliliters. Triclosan dismissal efficacy was measured by the given equation.
Elimination (%) = [(C_i_ − C_t_)/C_0_] × 100(2)

### 2.4. Kinetic Study

The kinetic study of triclosan removal by N-methyl butyl imidazolium bromide copper oxide nanocomposite was fixed at 5 to 40 min by taking a known quantity of N-methyl butyl imidazolium bromide copper oxide nanocomposite with triclosan solution in a water bath which was temperature controlled. The experiments were performed until the equilibrium was achieved. The amount of triclosan was determined at dissimilar time periods. The trials were completed in a similar method as for the sorption ones.

### 2.5. Thermodynamics Study

This study of triclosan was performed by using 20, 25 and 30 °C temperatures. The adjusted quantities of triclosan were utilized under these conditions.

### 2.6. Analysis of Triclosan by HPLC

The residual quantities of the triclosan were estimated by the HPLC procedure as given in the Appendix A. The mobile phase used was 2.5 pH acetic acid buffer-acetonitrile (30:70) with 1.0 mL/min flow rate. The column used was Sunniest RP Aqua C_28_ (25 cm × 4.6 mm id) and detection was achieved at 280 nm. The identification of triclosan was determined by equating the retention time of the standard with sample ones. The retention time of triclosan was 11.35 min.

### 2.7. Simulation Study

The simulation docking of triclosan uptake was performed as per the typical method given in Appendix A.

## 3. Results and Discussion

### 3.1. Characterization

N-methyl butyl imidazolium bromide copper oxide nanocomposite was characterized by FT-IR, which shows the presence of Cu-O in CuO. No peak was seen in the 600 to 610 region; confirming the non-availability of Cu_2_O [31]. The results are in agreement with Wu et al. [32] findings. The crystallinity of the prepared N-methyl butyl imidazolium bromide copper oxide nanoparticles was ascertained by XRD studies and confirmed monoclinic hexagonal crystalline CuO. The crystal size was calculated by the following formula.
D = Kλ/dcosθ
where, d, θ, K and D are full width at half maximum, reflection angle and constant, usually taken as 0.94. λ = 0.154 nm for Cu-Kα, β. The particle sizes calculated were from 50 to 98.5 nm.

The morphology of the prepared N-methyl butyl imidazolium bromide copper oxide nanoparticles was ascertained by SEM and the two photographs at 10,000 and 2,00,000 magnifications, showing the crystalline and rough surface. The shapes of the particles were irregular and spherical. Another study for characterization was TEM and the picture is shown in Figure 2, confirming the round and irregular shape of nanoparticles with sizes ranging from 50 to 99.5 nm. These results are in agreement with the XRD results. Based on the above argument, the following N-methyl butyl imidazolium bromide copper oxide nanoparticle structure (Figure 3) was developed, in which four molecules of N-methyl butyl imidazolium bromide are found on the exterior of CuO. This structure has a positive charge and a very good sorbent for negatively charged species.

### 3.2. Sorption Study

The uptake of triclosan was optimized by numerous parameters. The improved parameters were 20–120 µg/L triclosan; 0.1–1.5 g/L for N-methyl butyl imidazolium bromide copper oxide nanocomposite dosage; 3–12 medium pHs; time of 5–40 min with a temperature of 20–30 °C. The results of all these sets of experiments are given in the following sections.

#### 3.2.1. Concentration of Triclosan

The concentration of triclosan was optimized by taking 10 to 130 µg/L. The other variables controlled were contact time of 30 min, pH 8.0, dose 1.0 and 25 °C temperature. The results of these sets of experiments findings are graphed in Figure 4a and it is clear from this figure that the triclosan removal incased from 10 to 100 g/L concentration while further augment in the concentration could not result in more triclosan uptake. The sorption of triclosan were 9, 18.7, 28, 36, 45, 53.4, 63, 71.5, 80.5, 90.2, 90.3, 90.3 and 90.3 µg/g at 10, 20, 30, 40, 50, 60, 70, 80, 90, 100, 110, 120 and 130 µg/L concentrations. The percentage calculations were conducted and the percentage removal at equilibrium was 90.2. These data are a confirmation of the fact that 100 µg/L concentration was the optimized one at equilibrium. Therefore, 100 µg/L concentration of triclosan was considered as the best one in this set of experiments.

#### 3.2.2. Contact Time

The contact time for triclosan removal was optimized by taking 5 to 40 min. The other variables controlled were starting concentration 100 µg/L, pH 8.0, dose 1.0 and 25 °C temperature. The results of these sets of experiments findings are graphed in Figure 4b and it is clear from this figure that the triclosan removal increased from 5 to 30 min time while further augment in the contact time could not result in more triclosan uptake. The sorption of triclosan were 13.6, 27.9, 42.5, 57.15, 72.90, 90.2, 91.3 and 91.3 µg/g at 5, 10, 15, 20, 25, 30, 35, and 40 min. The percentage calculations were done and the percentage removals were 13.6, 27.9, 42.5, 57.15, 72.90, 90.2, 91.3 and 91.3. These data are a confirmation of the fact that 30 min of contact time was the optimized one at equilibrium. Therefore, 30 min contact time for triclosan was considered the best one in this set of experiments.

#### 3.2.3. pH of the Solution

The solution pH for triclosan removal was optimized by taking 3 to 12 units. The other variables controlled were starting concentration 100 µg/L, contact time 30 min, dose 1.0 and 25 °C temperature. The results of these sets of experiment findings are graphed in Figure 4c and it is clear from this figure that the triclosan removal increased from 3 to 8 pH while further augment in the pH could not result in more triclosan uptake. The sorptions of triclosan were 22.3, 32.4, 41.5, 57.4, 83.6, 91.2, 93.2, 93.3, 94.5 and 94.4 µg/g at 3, 4, 5, 6, 7, 8, 9, 10, 11 and 12 pH. The percentage calculations were done and the percentage removals were 22.3, 32.4, 41.5, 57.4, 83.6, 91.2, 93.2, 93.3, 94.5 and 94.4. These data are a confirmation of the fact that 8.0 pH was the optimized one at equilibrium. Therefore, 8.0 pH for triclosan was considered the best one in this set of experiments. It is important to emphasize that many natural water resources have 7–8 pH and, therefore, it may be concluded that the given method may be highly useful to natural water systems.

#### 3.2.4. Dose of the Nanocomposite

The nanocomposite dose for triclosan removal was optimized by taking 0.1, 0.25, 0.50, 0.75, 1.0, 1.25 and 1.5 g/L concentration. The other variables controlled were starting concentration 100 µg/L, contact time 30 min, pH 8.0 and 25 °C temperature. The results of these sets of experiment findings are graphed in Figure 4d and it is clear from this figure that the triclosan removal increased from 0.1 to 1.0 g/L while further augment in dose could not result in more triclosan uptake. The sorptions of triclosan were 35.6, 55.4, 72.8, 81.7, 91.2, 91.2 and 92.4 µg/g at 0.1, 0.25, 0.50, 0.75, 1.0, 1.25 and 1.50 g/L. The percentage calculations were done and the percentage removals were 35.6, 55.4, 72.8, 81.7, 91.2, 91.2 and 92.4. These data are a confirmation of the fact that 1.0 g/L dose was the optimized one at equilibrium. Therefore, 1.0 g/L dose for triclosan was considered as the best one in this set of experiments.

#### 3.2.5. Temperature of the Solution

The experimental temperature for triclosan removal was optimized by taking 20, 25 and 30 °C. The other variables controlled were starting concentration 100 µg/L, contact time 30 min, 1.0 g/L dose and pH 7.0. The results of these sets of experiment findings are graphed in Figure 4e and it is clear from this figure that the triclosan removal decreased from 20 to 30 °C while the experiments that further increased or decreased temperature were not conducted because these are not the temperatures of natural water resources. The sorptions of triclosan were 9.0, 18.7, 28.5, 37.0, 46.4, 54.7, 64.2, 72.8, 82.2, 91.5, 92.3 and 92.3 μg/g at 20 °C while these values at 25 °C were 9.0, 18.7, 28.0, 36, 45, 53.4, 63, 71.5, 80.5, 90.2, 90.3, and 90.3 μg/g. The values at 30 °C were 9.0, 18.7, 26.8, 35, 43.6, 51.5, 60.4, 70.1, 79.5, 88.5, 88.3 and 88.2 μg/g. These data are a confirmation of the fact that the sorption was in the order of 20 > 25 > 30 °C, confirming the exothermic sorption of triclosan on the reported nanocomposite.

### 3.3. Combined Effect of the 3D Parameters

pH is one of the most important factors for determining the applicability of a method in the treatment of water in real-life problems. Therefore, efforts were made to study the combined effect of pH with initial concentration, time, dose and temperature. The results are given in Figure 5. A look at Figure 5a indicates that the best removal was 90.2% at 100 µg/L concentration and pH 8. A further increase in both concentration and pH could not augment the removal of triclosan. Similarly, a look at Figure 5b indicates that the best removal was 90.0% at 30 min and pH 8. A further increase in both contact time and pH could not augment the removal of triclosan. A look at Figure 5c indicates that the best removal was 91.0% at a 1.0 g/L dose and pH 8. A further increase in both dose and pH could not augment the removal of triclosan. In addition, the effects of %removal vs. temperature vs. pH are shown in Figure 5d, which indicates that the best removal was 91.5% at 20 °C temperature and pH 8. A further increase in pH could not augment the sorption of triclosan. It is interesting to mention that the adsorption decreased at high temperatures. Therefore, it was decided to select 25 °C temperature throughout this study because this is the temperature of most of the water resources. Finally, 25 °C temperature was selected to carry out all the experiments and the maximum sorption of triclosan was 91.0% at pH 8. The experiments were conducted five times (*n* = 5) and the percentage errors were calculated. The percentage error values were in a range of 2.5 to 4.5. These values clearly showed the approval of the experiments.

### 3.4. Modeling

The data obtained after experiments were used to run the models and the most significant isotherms used were Langmuir, Freundlich, Temkin and Dubinin–Radushkevich. The findings after running these models are given below.

#### 3.4.1. Langmuir

The equation for the Langmuir model is given in the Appendix A and this equation was used to model the experimental data. The Langmuir model parameters were determined and expressed by b (L/µg) and X_m_ (µg/g). The magnitudes of these parameters were exploited by the intercept and slope of the plot drawn between 1/*Q*_t_ and 1/*C*_t_ (Figure 6a). These values are presented in Table 1, which confirmed 0.087, 0.015 and 0.008 µg/g values of X_max_ at 20, 25 and 30 °C, respectively, while the values of b were 188.68, 625 and 833.33 L/µg at these temperatures. The magnitude of regressing constants was 0.929, 0.967 and 0.901. 

#### 3.4.2. Freundlich

The equation for the Freundlich model is given in the Appendix A and this equation was used to model the experimental data. The Freundlich model parameters were determined and expressed n (µg/L) and k_F_ (µg/g) (Figure 6b). The magnitudes of these parameters were exploited by intercept and slope of the plot drawn between logQt and logCt. These values are presented in Table 1, which confirmed 17.64, 10.11 and 5.08 µg/g values of k_F_ at 20, 25 and 30 °C, respectively, while the values of n were 1.38, 1.09 and 0.89 L/µg at these temperatures. The magnitude of regressing constants was 0.949, 0.968 and 0.885. The outcome of the experimental data by Freundlich and Langmuir models was studied. A comparison can be seen in Table 1 and it is clear that the regression constants for Langmuir are close to unity in comparison to the Freundlich. It means that the experimental data followed the criterion of the Langmuir model rather than Freundlich one. Finally, it was concluded that the sorption process proceeded through the Langmuir model.

#### 3.4.3. Temkin

The equation of the Temkin model is given in the Appendix A and this equation was used to model the experimental data. The Temkin model parameters were determined and expressed by K_T_ (L/µg) B_T_ and (kJ/mol) constants. The magnitudes of these parameters were exploited by slope and intercept of the plot drawn between logQt and logCt (Figure 6c). These values are presented in Table 1, which confirmed 5.55, 2.99 and 1.86 L/µg of K_T_ at 20, 25 and 30 °C, respectively, while 3.35, 2.70 and 2.23 were the values of B_T_ at 20, 25 and 30 °C, respectively. The magnitude of regressing constants was 0.949, 0.968 and 0.885 at K_T_ at 20, 25 and 30 °C, respectively. These values are approaching unity, which is the confirmation of the validity of the data by the Temkin model.

#### 3.4.4. Dubinin–Radushkevich

The equation for the Dubinin–Radushkevich model is given in the Appendix A and this equation was used to model the experimental data. The Dubinin–Radushkevich model parameters were determined and expressed by Q_m_ (µg/g), K_ad_ (mol^2^/kJ^2^) and E (kJ/mol). The values of Q_m_ (µg/g) and E (kJ/mol) were determined by the intercept and slope of the plot drawn between the graph of 1/Q_t_ and 1/C_t_ (Figure 6d). The degrees of K_ad_ (mol^2^/kJ^2^) were computed by exploiting an equation as given below.
lnQ_e_ = lnQ_m_ − K_ad_·E^2^(3)
where Q_e_, Q_m_ and E have the usual meaning.

The values of different constants are presented in Table 1, which confirmed 128.83, 158.48 and 177.83 µg/g of Q_m_ at 20, 25 and 30 °C, respectively, while 1.22, 3.20 and 5.83 mol^2^/kJ^2^ were the values of K_ad_ at 20, 25 and 30 °C, respectively. The degrees of E were 0.53, 0.42 and 0.36 kJ/mol at 20, 25 and 30 °C, respectively. The magnitude of regressing constants was 0.901, 0.954 and 0.849 at K_T_ at 20, 25 and 30 °C, respectively. These values are approaching unity, which is the confirmation of the validity of the data by the Dubinin–Radushkevich model.

### 3.5. Kinetic Study

First, second and Elovich’s kinetic models were exploited for computing the kinetics of the triclosan sorption process. The equations used for these models are given in the SI (Table S2). First, second and Elovich’s kinetic models’ graphs are plotted between log(Q_e_-Q_t_), t vs. Qt and t, lnt vs. Qt, respectively. The slopes and intercepts of these plots were exploited to compute the degrees of the various constants. The values of these different constants are presented in Table 2. Taking the example of first order, the value of the rate constant was 0.073 1/min with 90.2 experimental and 124.11 theoretical values of the equilibrium concentration of triclosan. These values varied from each other by 33.91, confirming their closeness. The value of the regression constant was 0.995, approaching unity. Taking the example of second order, the values of rate constant were 3.26 × 10^−6^ g/µg min with 90.2 experimental value and 909.09 theoretical value. These values varied from each other by 818.89%, confirming their non-closeness. The value of h was 2.70 µg/g min, confirming the good value of the initial rate of the sorption. The value of the regression constant was 0.994. The data of the first- and second-pseudo-order reaction models were matched. It was observed that the experimental and theoretical degrees of the sorbed triclosan amount were in agreement in the first-pseudo-order reaction, rather than in the second-pseudo-order reaction model. Further, the degrees of the regression constants of triclosan were slightly higher in the case of first order in comparison to the second-order reaction. Consequently, it was decided that the investigational data obeyed the first-pseudo-kinetic reaction. The degrees of adsorption (α) and desorption (β) rates were computed by Elovich’s kinetic model and the computed values are presented in Table 2; it is clear that α was 9.37 µg/g min while β was 0.28 g/µg, confirming faster sorption than desorption. The degree of the regressing coefficient was 0.939, approaching unity. These data confirmed the applicability of Elovich’s kinetic model.

### 3.6. Thermodynamics Study

Triclosan sorption thermodynamics of the adsorption phenomenon was ascertained by the thermodynamics equations, which are given in the Appendix A. The thermodynamics were estimated in terms of entropy, enthalpy and free energy. The degrees of entropy and enthalpy were 0.47, −0.11 and −5.43 × 10^−2^, respectively. The degrees of free energy at 20, 25 and 30 °C were −53.18, −74.17 and −76.14, respectively. The negative values of enthalpy, free energy and entropy are an indication of impulsive sorption and exothermic process.

### 3.7. Adsorption Mechanism

In scientific discoveries, the mechanism of any phenomenon is very important and it is also true with this manuscript. The mechanism of triclosan sorption on N-methyl butyl imidazolium bromide copper oxide nanocomposite was studied by exploiting the liquid-film diffusion kinetic model and the intra-particle diffusion kinetic models (Figure 7a,b). The equations for the liquid-film diffusion kinetic model and the intra-particle diffusion kinetic models are given in the Appendix A and these equations were used to model the experimental data. The liquid-film diffusion kinetic model parameters were determined and expressed by the rate constant (1/min), intercept and regression constant. The magnitude of the rate constant was exploited by the slope of the plot drawn between t and ln(1−F). This value is presented in Table 3, which confirmed 0.073 1/min. The magnitude of regressing constant was 0.946 with 0.32 as the intercept value. This information shows that the graph is approaching its origin. Contrarily, the intra-particle diffusion kinetic model parameters were determined and expressed by the rate constant (k_ipd1_), intercept and regression constant. The magnitude of this constant was exploited by the slope of the plot drawn between Qt and t^0.5^ (Figure 7a,b). The values are presented in Table 3, which confirmed 23.37 k_ipd1_. The magnitude of regressing constant was 0.976 with −43.72 as the intercept value. This information shows that the graph is not going through the origin. By comparing both models, it was found that the graph of the liquid-film diffusion kinetic model is passing closer to the origin in comparison to the intra-particle diffusion kinetic model. Moreover, the value of the regression constant was greater in the former model in comparison to the latter. These findings are clear proof of the applicability of the liquid-film diffusion kinetic model in the present sorption process.

### 3.8. Supramolecular Mechanism of Uptake

In addition to using the sorption mechanism model, it is also interesting if the sorption mechanism is developed and this is possible by a simulation study and triclosan and N-methyl butyl imidazolium bromide ionic liquid chemistry. The procedure for simulation between triclosan and N-methyl butyl imidazolium bromide is given in the Appendix A. The outcomes of the binding were calculated in terms of bonding and residues involved. The model developed after simulation between triclosan and N-methyl butyl imidazolium bromide and by chemical methods is shown in Figure 8. This figure indicates bonding between the hydroxyl group of triclosan and imidazole ring of N-methyl butyl imidazolium bromide. The maximum adsorption was observed at pH 8.0 and it can be explained at the supramolecular level by considering the model presented in Figure 8. The pKa value of N-methyl butyl imidazolium bromide is 7.0–7.4 and it means that N-methyl butyl imidazolium bromide ionizes at pH 7.0–7.4, leading to the formation of a cation of N-methyl butyl imidazolium bromide. On the other hand, the pKa value of 7.9–8.1 means that triclosan ionizes at 7.9–8.1 pH, leading to the formation of the anion of triclosan. Finally, the cation and anion interacted by electrovalent bonding and formed a complex structure, as shown in Figure 8. Further, it is important to mention that the pKa value of triclosan is controlling sorption because the pKa value of triclosan is higher than the pKa value of N-methyl butyl imidazolium bromide. The residues involved in the formation of the complex structure are the phenoxide group of triclosan and imidazolium ring of N-methyl butyl imidazolium bromide. These interpretations are in a good arrangement for supporting the adsorption behavior of triclosan. Finally, the uptake mechanism at the supramolecular scale is developed and established.

### 3.9. Regeneration of Sorbent

The regeneration of any sorbent is one of the important issues in developing economic water treatment methods. The regeneration was tried with water of different pHs, buffers of different pHs, acids, bases and organic solvents. The maximum desorption (98.6%) was obtained with acetone and the efficiency of the regenerated sorbent was not good, as it could remove further triclosan to 73%. On the other hand, the ethanol could desorb 96.8% triclosan and the removal capacity of the sorbent was good, i.e., 90.0%, which decreased to 85.6% after five cycles of regeneration.

## 4. Conclusions

Copper oxide nanoparticles were synthesized using green methods and the nanocomposite was prepared by using N-methyl butyl imidazolium bromide ionic liquid. The nanocomposite was characterized by using proper spectroscopic methods. The nanocomposite was used to remove triclosan in water and the percentage removal was 90.2% at pH 8.0, which is an asset for using this method in natural water conditions. In addition, 30 min is the fast time to apply the method in natural situations. The conditions developed for this method were time 30 min, concentration 100 µg/L, pH 8.0, dose 1.0 g/L and 25 °C temperature. The results were verified by Langmuir isotherm. In addition, Temkin and and D-Rs models were followed well by the data. The sorption mechanism was through the liquid-film diffusion kinetic model. The enthalpy and free energy values were negative while the entropy of the value was positive, which confirmed the natural sorption phenomenon at all the working temperatures. In addition, these results also confirmed the exothermic elimination of triclosan. The supramolecular simulation study confirmed electrostatic interaction between triclosan and N-methyl butyl imidazolium bromide ionic liquid. This method is green in nature and economical in behavior, giving 90% elimination of triclosan. Finally, this method is considered worthwhile for the elimination of triclosan in water.

## Figures and Tables

**Figure 1 molecules-27-05358-f001:**
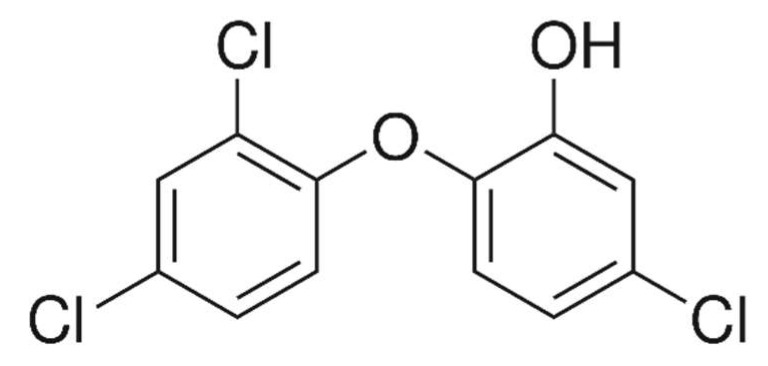
Structure of triclosan.

**Figure 2 molecules-27-05358-f002:**
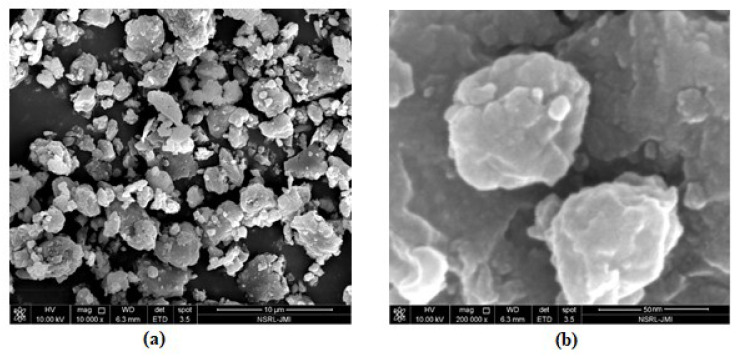
TEM images of spectrum of N-methyl butyl imidazolium bromide copper oxide nanocomposite: (**a**) at 10,000 and (**b**) 2,00,000 magnifications (ranging from 50 to 99.5 nm).

**Figure 3 molecules-27-05358-f003:**
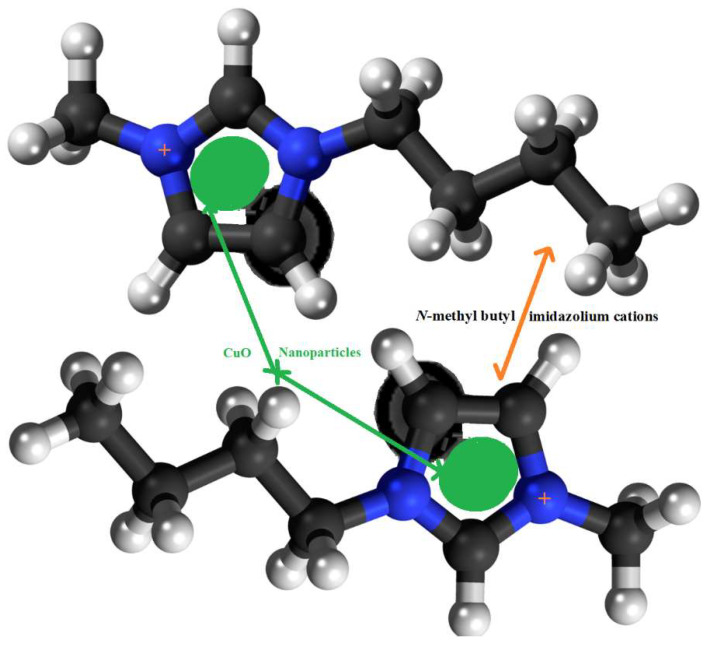
The structure of N-methyl butyl imidazolium cation copper oxide nanocomposite.

**Figure 4 molecules-27-05358-f004:**
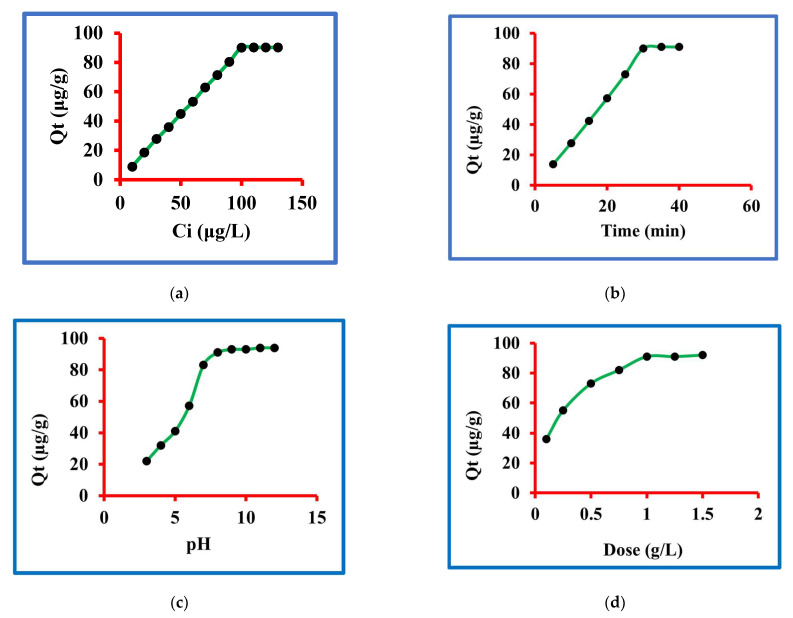
Sorption of triclosan (**a**) amounts, (**b**) time, (**c**) pH, (**d**) dose and (**e**) temperature.

**Figure 5 molecules-27-05358-f005:**
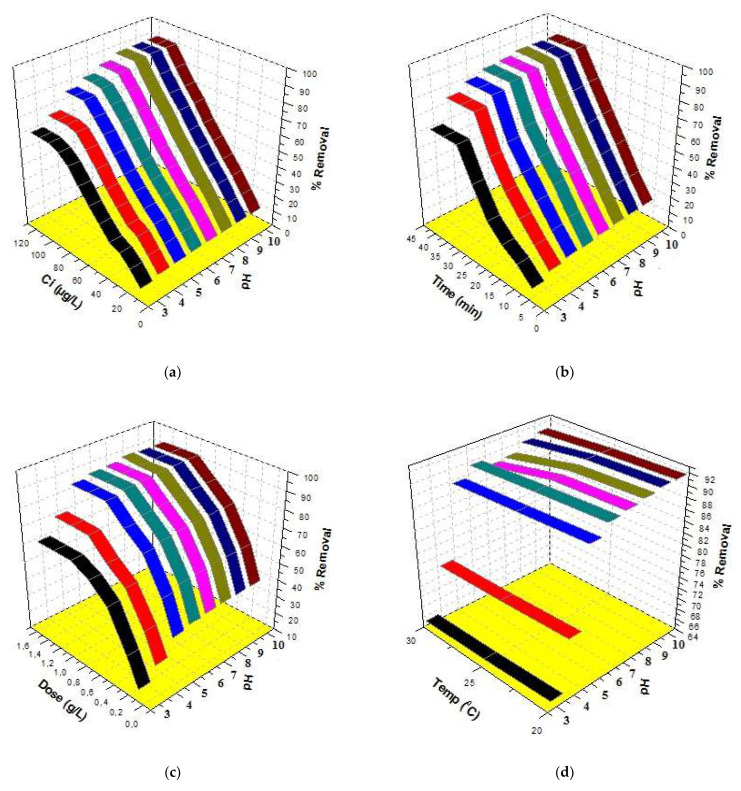
Three-dimensional plots for triclosan removal: (**a**) %removal vs. Ci vs. pH, (**b**) %removal vs. time vs. pH, (**c**) %removal vs. dose vs. pH and (**d**) %removal vs. temperature vs. pH.

**Figure 6 molecules-27-05358-f006:**
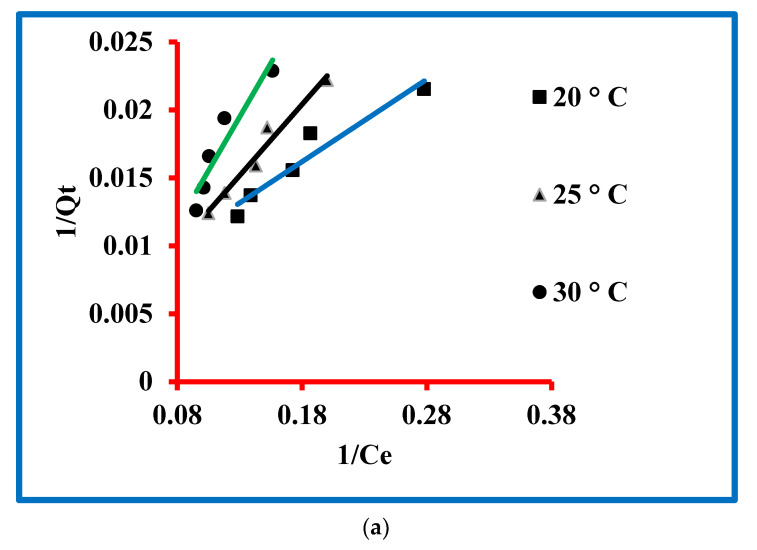
Models for triclosan: (**a**) Langmuir, (**b**) Freundlich, (**c**) Temkin and (**d**) Dubinin–Radushkevich.

**Figure 7 molecules-27-05358-f007:**
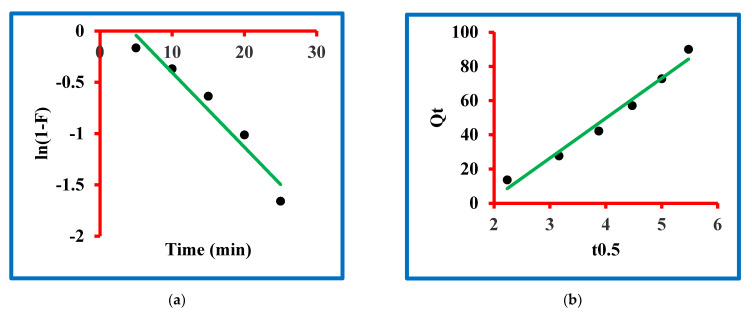
Mechanism of the sorption: (**a**) liquid-film diffusion and (**b**) intra-particular diffusion models.

**Figure 8 molecules-27-05358-f008:**
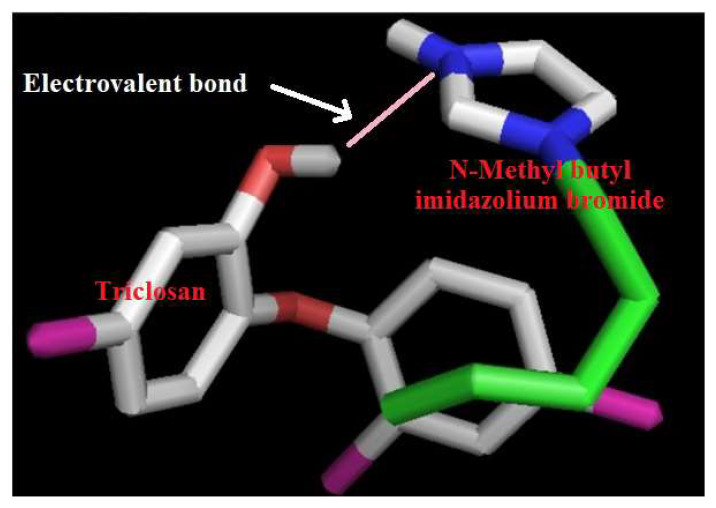
Interacting model of triclosan with N-methyl butyl imidazolium bromide.

**Table 1 molecules-27-05358-t001:** The adsorption isotherm parameters of triclosan.

Isotherms	Temperatures		
	20.0 °C	25.0 °C	30.0 °C
**Langmuir**			
X_max_ (μg/g)	0.087	0.015	0.008
b (L/μg)	188.68	625	833.33
R^2^	0.929	0.967	0.901
**Freundlich**			
k_F_ (µg/g)	17.64	10.11	5.08
n (µg/L)	1.38	1.09	0.89
R^2^	0.949	0.968	0.885
Temkin			
K_T_ (L/µg)	5.55	2.99	1.86
B_T_ (kJ/mol)	3.35	2.70	2.23
R^2^	0.949	0.968	0.885
**Dubinin–Radushkevich**			
Q_m_ (µg/g)	128.83	158.48	177.83
K_ad_ (mol^2^/kJ^2^)	1.22	3.20	5.38
E (kJ/mol)	0.53	0.42	0.36
R^2^	0.901	0.954	0.849

X_max_ and b = Langmuir; k_F_ and n = Freundlich constants; k_T_ and B_T_ = Temkin constants; Qm, Kad and E = Dubinin–Radushkevich constants and R^2^ = Regression coefficient.

**Table 2 molecules-27-05358-t002:** Kinetic parameters of triclosan.

Kinetic Models and Parameters	Numerical Values
**First-second-order**	
k_1_ (1/min)	0.073
The experimental Qe (µg/g)	90.2
The theoretical Qe (µg/g)	124.11
R^2^	0.995
**Pseudo-second-order**	
k_2_ (g/µg min)	3.26 × 10^−6^
The experimental Qe (µg/g)	90.2
The theoretical Qe (µg/g)	909.09
h (µg/g min)	2.70
R^2^	0.994
**Elovich**	
α (µg/g min)	9.37
β (g/µg)	0.028
R^2^	0.939

**Table 3 molecules-27-05358-t003:** The values of the uptake mechanism parameters of triclosan.

The Kinetic Models and Parameters	Numerical Values
**The intraparticle diffusion kinetic model**	
k_id_ (µg/g min^0.5^)	23.37
Intercept	−43.72
R^2^	0.976
**Liquid film diffusion kinetic model**	
k_fd_ (1/min)	0.073
Intercept	0.32
R^2^	0.946

## Data Availability

Data is contained within the article.

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
