# Peer review of "An Ionic-Liquid-Imprinted Nanocomposite Adsorbent: Simulation, Kinetics and Thermodynamic Studies of Triclosan Endocrine Disturbing Water Contaminant Removal"

_molecules, 2022, doi:10.3390/molecules27175358_

Round 1
Reviewer 1 Report
- Abstract: The authors should express maximum triclosan adsortion in terms of concentration (mg/L) under the better experimental conditions.
- Introduction – line 53: “molecular mass”
- Introduction: The authors should better explain why N-methyl butyl imidazolium bromide ionic 79 liquid-based copper oxide nanocomposite was prepared and used as potential adsorbent in triclosan removal. A variety of low-cost adsorbents, including agricultural wastes could also be used for such a purpose. The authors should add briefly technical information on the application of this material in adsorption studies.
- Results and discussion – lines 189-191; 202-204; 214-216; 228-230; 241-243: The discussion should be strongly revised. The influence of certain factors on the adsorption process should be better discussed. The present discussion is based only on the description of all experimental results. The present form is hard to read. The authors should explain that (suggestion) “a linear adsorption profile was observed varying from 10 to 100 microgram per litre of triclosan, followed by a stabilization (support saturation). Maximum adsorption capacity was XX” for Fig. 4a. The same for Fig. 4b,c,d. The authors should better explain the experimental results and remove this excessive number of experimental data in the text. These values of maximum adsorption for each assay should be reported in a Supplementary Material. I suggest a complete revision of 3.2. Sorption study Section.
- Results and discussion – Section 3.4: The authors should report some technical information for each isotherm model used for describing the adsorption process. The discussion must be strongy improved. In this version, it lacks a deep technical analysis (discussion) of the results.
- Results and discussion – Section 3.5: The authors should report some technical information for each kinetic model used for describing the adsorption process. The discussion must also be strongy improved. In this version, it lacks a deep technical analysis of the results.
- Results and discussion: Some studies have proved that non-linear adsorption models are highly advisable. The authors should fit non-linear kinetic/isotherm models to the experimental data. According to Tan and Hameed (2017), a large experimental error can make the right model fit poorly and the wrong model fit adequately, thereby providing misleading kinetic information. Fortunately, the nonlinear forms of kinetic models are more robust toward experimental errors and are hence preferred, especially when experimental errors are not controlled. See some references. I recommend cite such references:
https://doi.org/10.1016/j.cej.2009.09.013
https://doi.org/10.1016/j.jtice.2017.01.024
https://doi.org/10.1016/j.ijbiomac.2018.08.190
https://doi.org/10.1016/j.enzmictec.2019.05.001
https://doi.org/10.1007/s42452-019-0977-3
https://doi.org/10.1007/s42452-019-0977-3
- Results and discussion: Several error functions have been succesfully used as fittingness tests by measuring the difference between experimental and model-calculated equilibrium adsorption data. The authors should calculate errors for all analyses.
- Results and discussion: Desorption and regeneration studies (influence of several desorption reactants; influence of desorption agent concentration and reusability tests) shoul be incorporated to this study. It is more interesting to the readers.
- Results and discussion: Possible mechanisms of interaction between triclosan and adsorbent (functional groups on its surface) should also be explained in this study.
Author Response
Pointwise replies
Manuscript ID: molecules-1842638
Title: An ionic liquid imprinted nanocomposite adsorbent: Simulation,
kinetics and thermodynamic studies of triclosan endocrine disturbing water
contaminant removal
First of all, I would like to thank Ms. Jovana Curic, Section Managing Editor, to give us a chance for revising this manuscript. Besides, thanks are also the scholarly reviewer to give fruitful suggestions. Really, the incorporation of all the suggestions made this manuscript more useful and attractive to the readers. The point-wise replies to the comments of reviewers are given below.
Comments and Suggestions for Authors
Reply:
Thanks to this reviewer for his/her appreciation of our work.
Also, thanks for sparing his/her valuable time reviewing this manuscript and giving fruitful suggestions.
- Abstract: The authors should express maximum triclosan adsortion in terms of concentration (mg/L) under the better experimental conditions.
Reply:
It is provided as suggested.
- Introduction – line 53: “molecular mass”
Reply:
It is corrected as suggested.
- Introduction: The authors should better explain why N-methyl butyl imidazolium bromide ionic 79 liquid-based copper oxide nanocomposite was prepared and used as potential adsorbent in triclosan removal. A variety of low-cost adsorbents, including agricultural wastes could also be used for such a purpose. The authors should add briefly technical information on the application of this material in adsorption studies.
Reply:
About a decade ago scientists were using adsorbents developed from many raw materials but nowadays, researchers have started to use nanomaterials due to their remarkable properties. Besides, the nanocomposites have organic moieties making nanomaterials more effective in water treatment. Please refer to Figure 9 and you will get the idea why this adsorpbent was used. The reason is this that the ionic liquid parts have tendencies to bind triclosan more effectively in comparison to other sorbents.
- Results and discussion – lines 189-191; 202-204; 214-216; 228-230; 241-243: The discussion should be strongly revised. The influence of certain factors on the adsorption process should be better discussed. The present discussion is based only on the description of all experimental results. The present form is hard to read. The authors should explain that (suggestion) “a linear adsorption profile was observed varying from 10 to 100 microgram per litre of triclosan, followed by a stabilization (support saturation). Maximum adsorption capacity was XX” for Fig. 4a. The same for Fig. 4b,c,d. The authors should better EXPLAIN THE EXPERIMENTAL RESULTS and remove this excessive number of experimental data in the text. These values of maximum adsorption for each assay should be reported in a Supplementary Material. I suggest a complete revision of 3.2. Sorption study Section.
Reply:
As suggested by the learned reviwer, we have EXPLAINED THE EXPERIMENTAL RESULTS. We have published more than 100 papers in this area and using the same trend of the discussion. If this reviewer has some novel thing in mind we request to kindly suggest the exact changes to be needed.
- Results and discussion – Section 3.4: The authors should report some technical information for each isotherm model used for describing the adsorption process. The discussion must be strongy improved. In this version, it lacks a deep technical analysis (discussion) of the results.
Reply:
Section 3.4 deals with the results obtained by analysing the experimental data with various models. This is the practice all over the world. All researchers are using the same pattern. . If this reviewer has something special in mind we request to write what exactly he needs to suggest.
- Results and discussion – Section 3.5: The authors should report some technical information for each kinetic model used for describing the adsorption process. The discussion must also be strongy improved. In this version, it lacks a deep technical analysis of the results.
Reply:
Section 3.5 deals with the kinetics results by applying various models. From the results, it was determined that sorption followed a first-pseudo-order reaction model. It is the standard practice to determine. If this reviewer has something special in mind we request to write what exactly he needs to suggest.
- Results and discussion: Some studies have proved that non-linear adsorption models are highly advisable. The authors should fit non-linear kinetic/isotherm models to the experimental data. According to Tan and Hameed (2017), a large experimental error can make the right model fit poorly and the wrong model fit adequately, thereby providing misleading kinetic information. Fortunately, the nonlinear forms of kinetic models are more robust toward experimental errors and are hence preferred, especially when experimental errors are not controlled. See some references. I recommend cite such references:
https://doi.org/10.1016/j.cej.2009.09.013
https://doi.org/10.1016/j.jtice.2017.01.024
https://doi.org/10.1016/j.ijbiomac.2018.08.190
https://doi.org/10.1016/j.enzmictec.2019.05.001
https://doi.org/10.1007/s42452-019-0977-3
https://doi.org/10.1007/s42452-019-0977-3
Reply:
We have published many papers in this area and are aware of this point. We tried the non-linear form but the best results are described by the linear form of models.
- Results and discussion: Several error functions have been succesfully used as fittingness tests by measuring the difference between experimental and model-calculated equilibrium adsorption data. The authors should calculate errors for all analyses.
Reply:
The error data is given in the revised manuscript.
- Results and discussion: Desorption and regeneration studies (influence of several desorption reactants; influence of desorption agent concentration and reusability tests) shoul be incorporated to this study. It is more interesting to the readers.
Reply:
The desorption and regeneration are given in the revised manuscript.
- Results and discussion: Possible mechanisms of interaction between triclosan and adsorbent (functional groups on its surface) should also be explained in this study.
Reply:
The possible mechanisms of interactions between triclosan and adsorbent (functional groups on its surface) are already described by modeling approach. Please see section 3.8 and Figure 9.

Reviewer 2 Report
1. The manuscript needs a sample table, which describes the source of all compounds used, their initial purity, purification methods, and final purity of the samples as used.
2. What’s the reason for the select of temperature for the study of triclosan.
3. Please adjust the chart format according to the requirements of the journal.
4. All symbols used in the table must be defined in the table heading/footnote.
5. FTIR and XRD figures should be added in the text. Figure 2 is the image of SEM, but line 167 said that Figure 2 is TEM, where is the TEM?
6. Line 191, pay attention to the details of the writing. Double” and”.
7. Did the authors explore the time or pH first?
Why is the pH of the 3.2.1 part of the experiment 7, and the pH of the 3.2.2 Contact time part changed to 8? That's not very reasonable.
In addition, lines 201-203, "The sorption of triclosan were 13.6, 27.9, 42.5, 57.15, 72.90, 90.2, 91.3 and 91.3 µg/g at 5, 10, 15, 20, 25, 30, and 40 minutes." I think you must have forgotten to write 35 minutes!
8. How many sets of parallel experiments were done, and why is there no error bar in the figure?
9. What is the maximum adsorption capacity calculated from the Langmuir isotherm adsorption isotherm, and does it have an advantage over other similar nanomaterials? What are the advantages? It should be listed in a table.
10. In section 2.1, CuO nanoparticles and N-methyl butyl imidazolium bromide were stirred for a further 2-3 hrs with constant heating. What is this heating method (water bath or oil bath)?
11. In section 2.1, what is the yield of the final sample?
12. The average particle sizes of samples should be given in Figure 2.
Author Response
Pointwise replies
Manuscript ID: molecules-1842638
Title: An ionic liquid imprinted nanocomposite adsorbent: Simulation,
kinetics and thermodynamic studies of triclosan endocrine disturbing water
contaminant removal
First of all, I would like to thank Ms. Jovana Curic, Section Managing Editor, to give us a chance for revising this manuscript. Besides, thanks are also the scholarly reviewer to give fruitful suggestions. Really, the incorporation of all the suggestions made this manuscript more useful and attractive to the readers. The point-wise replies to the comments of reviewers are given below.
Comments and Suggestions for Authors
Reply:
Thanks to this reviewer for his/her appreciation of our work.
Also, thanks for sparing his/her valuable time reviewing this manuscript and giving fruitful suggestions.
- The manuscript needs a sample table, which describes the source of all compounds used, their initial purity, purification methods, and final purity of the samples as used.
Reply:
We have used only one compound tricosan. It was obtained from Sigma-Aldrich Chemical Co. USA with 99% purity.
- What’s the reason for the select of temperature for the study of triclosan.
Reply:
It is the standard practice to study adsorption to calculate the thermodynamic properties of the adsorption process.
- Please adjust the chart format according to the requirements of the journal.
Reply:
The publisher has already set the manuscript as per their requirements.
- All symbols used in the table must be defined in the table heading/footnote.
Reply:
It is provided as suggested.
- FTIR and XRD figures should be added in the text. Figure 2 is the image of SEM, but line 167 said that Figure 2 is TEM, where is the TEM?
Reply:
It was typing mistake and corrected.
- Line 191, pay attention to the details of the writing. Double” and”.
Reply:
It is taken care.
- Did the authors explore the time or pH first?
Reply:
We optimized first pH.
Why is the pH of the 3.2.1 part of the experiment 7, and the pH of the 3.2.2 Contact time part changed to 8? That's not very reasonable.
Reply:
It was typing mistake, and corrected in the revised manuscript.
In addition, lines 201-203, "The sorption of triclosan were 13.6, 27.9, 42.5, 57.15, 72.90, 90.2, 91.3 and 91.3 µg/g at 5, 10, 15, 20, 25, 30, and 40 minutes." I think you must have forgotten to write 35 minutes!
Reply:
Yes dear Professor!!!
It was typing mistake, and corrected in the revised manuscript.
- How many sets of parallel experiments were done, and why is there no error bar in the figure?
Reply:
We repeated 5 sets of experiments and the percent errors are given in the text. Also another reviwer suggested to give errors in the text.
- What is the maximum adsorption capacity calculated from the Langmuir isotherm adsorption isotherm, and does it have an advantage over other similar nanomaterials? What are the advantages? It should be listed in a table.
Reply:
These were 0.087, 0.015. And 0.008 µg/g at 20, 25 and 30 °C temperatures. These results can not be compared because we could not find any paper describing sorption at these temperatures.
- In section 2.1, CuO nanoparticles and N-methyl butyl imidazolium bromide were stirred for a further 2-3 hrs with constant heating. What is this heating method (water bath or oil bath)?
Reply:
This section is edited and now more clear. Basically, the mixture of CuO nanoparticles and N-methyl butyl imidazolium bromide in a beaker was stirred and heated on a magnetic stirrer.
- In section 2.1, what is the yield of the final sample?
Reply:
The yield was 99.5%.
- The average particle sizes of samples should be given in Figure 2.
Reply:
It is given as suggested.

Round 2
Reviewer 1 Report
Authors have tried to answer my queries, however many points are not solved. That way, the paper still needs further revisions, please, recheck my previous report.
Author Response
Pointwise replies
Manuscript ID: molecules-1842638
Title: An ionic liquid imprinted nanocomposite adsorbent: Simulation,
kinetics and thermodynamic studies of triclosan endocrine disturbing water
contaminant removal
First of all, I would like to thank Ms. Jovana Curic, Section Managing Editor, to give us a chance for revising this manuscript. Besides, thanks are also the scholarly reviewer to give fruitful suggestions. Really, the incorporation of all the suggestions made this manuscript more useful and attractive to the readers. The point-wise replies to the comments of reviewers are given below.
Comments and Suggestions for Authors
Open Review
( ) I would not like to sign my review report
(x) I would like to sign my review report
English language and style
( ) Extensive editing of English language and style required
( ) Moderate English changes required
(x) English language and style are fine/minor spell check required
( ) I don't feel qualified to judge about the English language and style
Yes |
Can be improved |
Must be improved |
Not applicable |
|
Does the introduction provide sufficient background and include all relevant references? |
(x) |
( ) |
( ) |
( ) |
Are all the cited references relevant to the research? |
( ) |
(x) |
( ) |
( ) |
Is the research design appropriate? |
(x) |
( ) |
( ) |
( ) |
Are the methods adequately described? |
( ) |
( ) |
( ) |
( ) |
Are the results clearly presented? |
( ) |
(x) |
( ) |
( ) |
Are the conclusions supported by the results? |
( ) |
(x) |
( ) |
( ) |
Comments and Suggestions for Authors
Authors have tried to answer my queries, however many points are not solved. That way, the paper still needs further revisions, please, recheck my previous report.
Reply:
Thanks to this reviewer for his/her appreciation of our work.
Also, thanks for sparing his/her valuable time reviewing this manuscript and giving fruitful suggestions.
Reply:
We have revised the manuscript as per the reviewers comments but those comments were not understood by us which were general. That’s why we asked the exact comments from the reviewer.
Consequently the thank Ms. Jovana Curic, Section Managing Editor wrote this reviewer to give exact comments and this reviewer provided comments to thank Ms. Jovana Curic, Section Managing Editor, which she passed us as below.
Several suggestions were not performed, as can be observed in Response Letter. I recommended a deep discussion of the experimental results. However, the authors declare "We have published more than 100 papers in this area and using the same trend of the discussion. If this reviewer has some novel thing in mind we request to kindly suggest the exact changes to be needed". In fact, the authors kept the original discussion. The suggestions were recommend for improving the quality of the manuscript. Thus, I recommend the acceptance of this revised version.
Reply,
Thanks for accepting our manuscript. However, we have edited and revised the results and discussion once again.

Reviewer 2 Report
Thank you for your reply.
Author Response
Pointwise replies
Manuscript ID: molecules-1842638
Title: An ionic liquid imprinted nanocomposite adsorbent: Simulation,
kinetics and thermodynamic studies of triclosan endocrine disturbing water
contaminant removal
First of all, I would like to thank Ms. Jovana Curic, Section Managing Editor, to give us a chance for revising this manuscript. Besides, thanks are also the scholarly reviewer to give fruitful suggestions. Really, the incorporation of all the suggestions made this manuscript more useful and attractive to the readers. The point-wise replies to the comments of reviewers are given below.
Comments and Suggestions for Authors
Open Review
(x) I would not like to sign my review report
( ) I would like to sign my review report
English language and style
( ) Extensive editing of English language and style required
( ) Moderate English changes required
(x) English language and style are fine/minor spell check required
( ) I don't feel qualified to judge about the English language and style
Comments and Suggestions for Authors
Thank you for your reply.
Reply:
Thanks to this reviewer for his/her appreciation of our work.
Also, thanks for sparing his/her valuable time reviewing this manuscript and giving fruitful suggestions.
Also, thanks for accepting our manuscript.
